# Calculating worldwide needs for morphine for pain in advanced cancer and proportions feasibly met by country estimates of requirements and consumption. Retrospective, time-series analysis (1997–2017)

**Joseph Clark** [1]*, **Lucia Crowther** [1], **Miriam J. Johnson**[1], **Christina Ramsenthaler**[1,2], **David C. Currow**[3]

1 Wolfson Palliative Care Research Centre, University of Hull, Hull, United Kingdom, 2 Faculty of Health Sciences, Zurich University of Applied Sciences, Winterthur, Switzerland, 3 Faculty of Science, Medicine and Health, University of Wollongong, Wollongong, Australia

☯ These authors contributed equally to this work.
* joseph.clark@hyms.ac.uk

## Abstract

Lack of access to therapeutic opioids continuing causes global health inequalities. Access to morphine for symptom control is regulated under the terms of the Single Convention on Narcotics, countries must submit annual morphine requirement estimates and report consumption to the International Narcotics Control Board (INCB). INCB indicates access to morphine is increasing, however, estimated needs are unreported so changing proportions of needs feasibly met by requirements and consumption are unknown. Retrospective time series-analysis taking cross-sections every five years of gaps between *calculated needs* for morphine for people who die from cancer and *total treatable* using *estimates of requirements* and *consumption* (1997, 2002, 2007, 2012, 2017). We *calculated need* using INCB-recommended methods (80% of people who die from cancer require 67.5mg of morphine daily for 90 days (6.075g)) for countries reporting *estimates* and *consumption* using Global Burden of Disease cancer deaths by country. Gaps between *calculated need* and *total treatable* population using *estimates* and *consumption* were calculated. We report proportions of need feasibly met by *estimates* and *consumption* for included countries, by World Bank Income group. Global availability of morphine increased, from *estimates* sufficient to treat 86% of *calculated needs* in 1997, to 701% in 2017. However, proportion of countries *estimating requirement*s feasibly meeting >100% of *calculated needs* rose only from 16% to 30%. Almost all Low-and-Middle-Income Countries submitted inadequate *estimates* with little change in 20 years. *Consumption* was lower than *calculated needs* at all time-points. Very few countries reported *consumption* greater than their *estimate of requirement*. Most countries submitted morphine *estimates* insufficient to meet analgesic needs of people who died from cancer. *Estimates of requirements* contextualise future *Consumption*, and

**Data Availability Statement:** All data used are available in the public domain using references provided in the paper.

**Funding:** The authors received no specific funding for this work.

**Competing interests:** The authors have declared that no competing interests exist.

increases in adequacy of *estimates* and *consumption* were minimal over 20 years. Annual publication of *calculated morphine needs* alongside *estimates* and *consumption* may be a key step to drive countries' accountabilities.

## Introduction

Morphine is listed as an essential medicine by the World Health Organization (WHO) and is a key therapy in the WHO 3-step pain ladder [1, 2]. Differences in the availability and utilisation of morphine for people with pain and advanced cancer generate key global health inequalities [3]. In 2018, 79% of the world's population, living mainly in low- and middle-income countries, consumed only 13% of the world's consumption of licit morphine used for the management of pain [4]. Access to appropriate pain relief is acknowledged as a human right [5, 6].

In addition to well established clinical indications, opioids including morphine may be used illicitly and can be highly addictive [7]. For complex socioeconomic reasons, over-consumption in some countries–primarily the United States–has caused significant suffering from addiction and through diversion of licit opioids to illicit channels since the 1990s [8]. Appropriate regulation of morphine must, therefore, achieve balance between availability for beneficial therapeutic purposes and restriction of illicit use.

Since 1961, morphine has been internationally regulated under the terms of the Single Convention on Narcotics, implementation of which is overseen by the International Narcotics Control Board (INCB) [9]. The INCB is charged with prohibition of illicit production and supply of controlled substances, whilst ensuring use for medical treatment through use of the *Estimates System* [10].

Under the *Estimates System*, each year countries submit *Estimates of Requirements* for controlled substances to the INCB which set an upper limit on stock that can be held and the amount that can be traded across borders. *Estimates of requirements* are calculated quantities of controlled substances that countries can safely administer, <u>not</u> crude estimates of how much would be necessary to meet actual clinical needs. Gaps between calculated need and levels of calculated need feasibly met by *estimates of requirements* are not reported.

Flexibilities in global regulation are present, which allow for additional trade and consumption in unforeseen circumstances [11]. However, it is unclear whether countries avail themselves of this flexibility to increase consumption or if requirements determine consumption in practice.

To calculate *Estimates of requirements*, the INCB endorses three approaches [Box 1].

---

### Box 1. Methods recommended by the INCB for the quantification of requirements for controlled substances

Method A. Consumption-based methods and variants

"The consumption-based method and its variants are based on past health-care demands for controlled substances. Where past use of controlled substances is stable, future requirements can be estimated by averaging the amounts consumed in recent years and adding a margin for unforeseeable increases [p.21]".

Method B. Service-based method

---

"The service-based method starts by taking the quantities of controlled substances currently in use in standard health-care facilities and extrapolating those findings to similar facilities throughout the country [p.23]".

Method C. Morbidity-based method

"The morbidity-based method uses data on the frequency of health problems (morbidity) and an assumption of how those health problems will be treated (average standard treatment schedules) to calculate the requirements for controlled substances [p.26]".

Source: International Narcotics Control Board. Guide on Estimating Requirements for Substances Under International Control, 2012. Available from: www.incb.org

Methods A and B calculate future *estimates of requirements* based upon historic usage, which in context of chronic under-consumption (because of a lack of availability) would perpetuate a cycle of under-availability [12]. However, in 2012, to support use of the 'Morbidity-based method,' the INCB published a recommended formula for calculating *estimated requirements* for morphine for people in pain with advanced cancer which appears responsive to changing clinical needs [13].

The INCB suggests that 80% of people with advanced cancer will require morphine, at an average of 67.5 mg per day in the last 90 days of life. Using numbers of people who die from cancer by country, it is therefore possible to calculate overall morphine needs for people with pain from advanced cancer by country using the INCB-endorsed approach.

In addition to *Estimates of Requirements*, countries submit their annual Consumption of controlled substances to the INCB. Methods with which individual countries calculate *estimates of requirements* are unreported. However, at the service-level, reporting of consumption to national regulators of controlled substances is a prerequisite to obtaining and maintaining a license to prescribe opioids. A controlled substance is regarded as "consumed" when it has been supplied to any person or enterprise for retail distribution, medical use or scientific research; and "consumption" is construed accordingly [4].

The Lancet Commission on Pain and Palliative Care applied INCB assumptions measuring the difference between palliative care-estimated needs and the quantity available for prescription for patients [3]. These calculations broadly demonstrated gaps between need and availability worldwide, and highlighted that Human Development Index ratings are associated with huge inequities in adequacy of clinical care provision.

No other study has investigated the adequacy of countries' *estimates of requirements* nor described changes over time to explore the extent to which *estimates of requirements* contextualise future consumption. Adequate *estimates of requirements* would not ensure access to morphine for those with clinical need. However, INCB reports that "obtaining accurate information about the legitimate requirements for [controlled] substances is a prerequisite to ensuring their availability [p.iv, 2012] [13]." By contrast, overestimation of requirements relative to calculated needs increases the risks of opioid abuse and considerable impacts on the health and wellbeing of society.

We aimed to calculate needs for morphine for people who die from cancer using an adapted morbidity-based method [Box 1] and identify changes in proportions of calculated need which could feasibly be met by countries' submitted *estimates of requirements* using INCB-

recommended methods of calculation between 1997 and 2017 We report changes in *consumption* at five year intervals, to account for any consumption of morphine which is additional to *estimates of requirements*.

## Aims

- To calculate total needs for morphine using INCB's assumption that 80% of people who die from cancer will require 67.5 mg of morphine per day in the last 90 days of life

- To describe changes in the proportions of people in pain with advanced cancer feasibly treatable using existing data on morphine *estimates of requirements* and *consumption* reported to the INCB, by World Bank Income group over time (1997–2017).

## Methods

### Key terms

Our study uses a number of related but distinct key terms throughout the manuscript. Key terms are presented and defined in Box 2:

---

### Box 2. Definitions of key terms

- *Estimates system*–refers to the INCB's method of implementation of the Single Convention on Narcotics, which requires countries to submit Estimates of Requirements for controlled substances annually.

- *Estimates of requirements*–refers to estimated or calculated quantities of controlled substances individual countries submit to the INCB annually which can be safely administered for clinical or scientific purposes.

- *Consumption*–refers to quantities of controlled substances reported annually to the INCB as having been 'consumed.' Controlled substances are considered to have been 'consumed' where it has been supplied to any person or enterprise for retail distribution, medical use or scientific research

- *Calculated need*–refers to calculations of country-level need for morphine for people with advanced cancer at INCB-recommended dosage and duration used in this study (80% of people who die from cancer, 67.5mg per day for the last 90 days of life)

- *Total/feasibly treatable*–refers to total or proportion of *Calculated need* feasibly met using *Estimates of requirements* or *Consumption* assuming that all morphine is administered to people who die from cancer at INCB-recommended dosage and duration.

*Estimates of requirements are not equivalent to country estimates of clinical needs. The INCB notes that "in an ideal system, the requirements for controlled substances would equal the needs [INCB, p6]. [4]"

---

## Study design

This is a retrospective time series analysis taking repeated cross-sections every five years of gaps between *calculated needs* for morphine for people who die from advanced cancer and *total treatable* using *estimates of requirements* and countries' reports of *consumption*. We apply INCB guidance for calculating *estimates of requirements* for morphine for people who die from cancer to calculate how many people could feasibly be treated with countries' *estimates of requirements* and reported *consumption*. Our novel approach uses real-world data to apply global guidance and is therefore not reported in accordance with standard reporting checklists (e.g. STROBE, GATHER). For example, we provide crude calculations of need for morphine using total deaths and standard dosage/duration of morphine per patient, for which it is not necessary to evaluate uncertainty within the calculations.

## Data collection

Total deaths from cancer and country population data were retrieved for 195 countries for the years: 1997, 2002, 2007, 2012 and 2017 using Global Burden of Disease (GBD) and Our World in Data [14, 15].

Country *estimates of requirements* for morphine and reports of morphine *consumption* were then retrieved from the INCB Narcotic Drug–Technical Reports for countries for which we had data for cancer deaths. Data for 2002, 2007, 2012 and 2017 were available from INCB's website [16]. Further data regarding *estimates of requirements* and *consumption* data for 1997 were provided by the INCB on request. The INCB does not publish guidance on how to calculate *estimates of requirements* for other controlled substances which are used to treat pain in advanced cancer (e.g. codeine, fentanyl). Therefore, these are not included in our analysis. Morphine-equivalent doses were not calculated.

INCB reports of country *consumption* were altered from kilos into grams (consumption (kg) * 1000). For countries reporting "<1kg" consumption, we coded this as zero, as consumption <1kg is highly unlikely to have been systematically directed towards meeting the needs of people in pain from cancer.

Finally, we applied World Bank Income Group rankings for each included country for the relevant year to complete our dataset (1 = Low-income Country, 2 = Lower Middle-Income Country, 3 = Upper Middle-Income Country, 4 = High-income country) [17].

All data used in our study are aggregated data available in the public domain and we did not require ethical approval.

## Sample size and eligibility criteria

We aimed to include all countries in the world in our analysis which report *estimates of requirements* to the INCB. However, several countries were excluded for each year in the study due to different conceptualisation of countries between GBD and INCB where *estimates of requirements* were not reported to INCB for a given year [Fig 1].

Our repeated cross-sectional study includes different countries at each time point. However, we report proportions of *calculated need* met be feasibly met by *estimates of requirements* at each time-point which allows direct comparisons. Analysis of proportions of *calculated need* feasibly met be country reports of morphine *consumption* includes countries which reported *estimates of requirements* **and** consumption, which further reduced the sample size at each time-point [Fig 1].

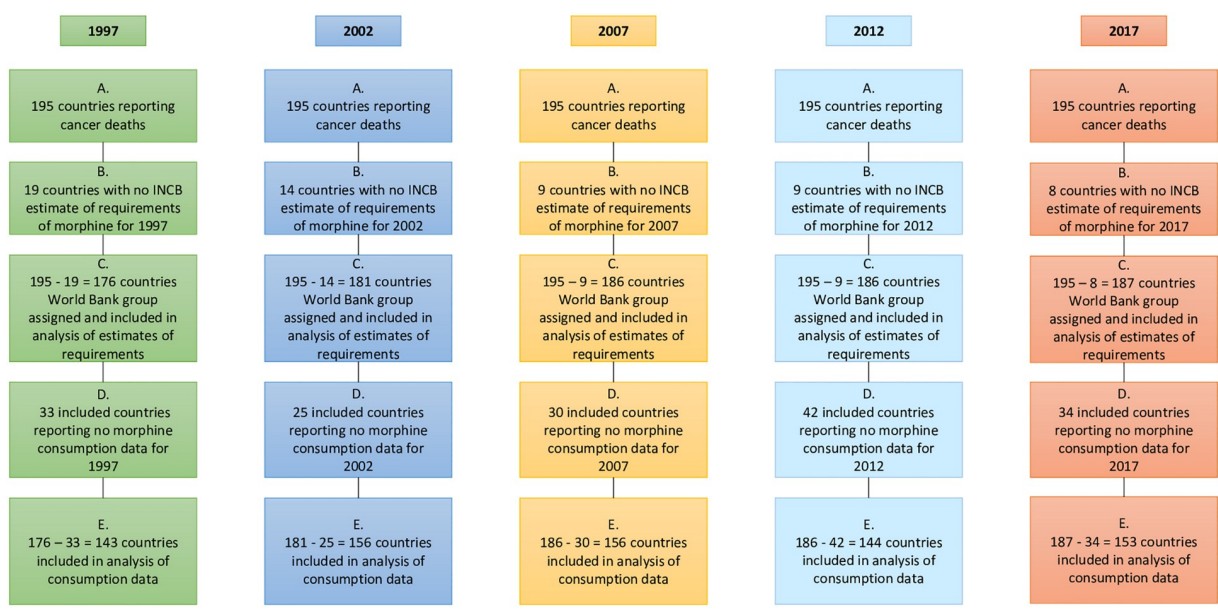

**Fig 1. Flow chart of total countries included/excluded at each stage of analysis.**

## Data analysis

We applied INCB-assumptions that 80% of people with late-stage cancer require morphine at an average dose of 67.5 mg per day for 90 days (6.075 g) [13]. Applying this assumption to proportion of cancer deaths for each country, we calculated worldwide needs for morphine treatment of people who died from cancer and calculated the proportions of need feasibly met by countries' *estimates of requirements* at INCB-recommended dosage and duration (1997–2017). To determine the extent to which *estimates of requirements* contextualise *consumption*, this method was applied to reports of morphine *consumption* to account for any *consumption* which is additional to that feasible from *estimates of requirements*. We conducted our analysis in three stages, summarised in Fig 2. Results are presented overall, by income group and by percentiles of need feasibly met by *estimates of requirements* and *consumption* [18].

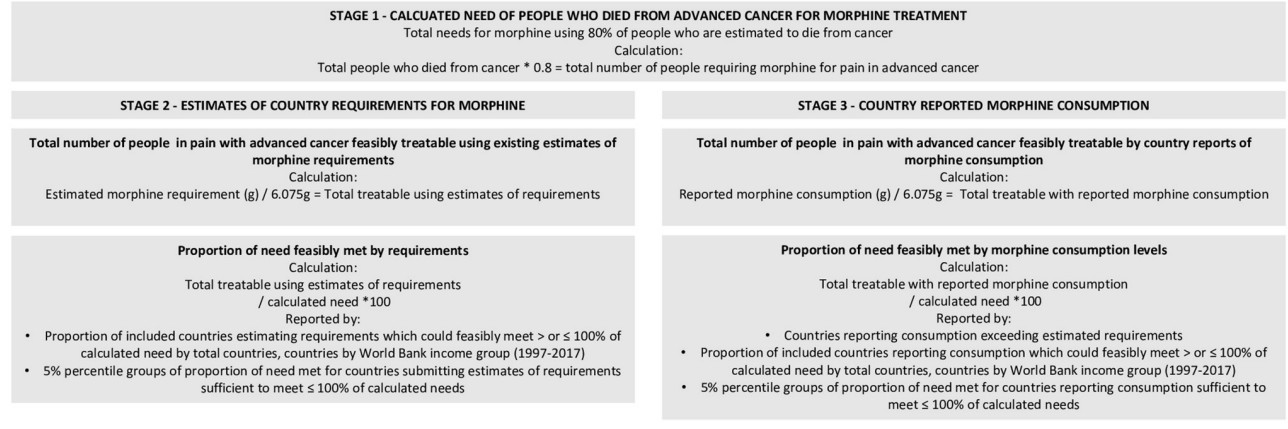

**Fig 2. Process of calculating needs and proportions of needs feasibly met** *by estimates of requirements* and *consumption*.

Data were managed and analysed using Microsoft Excel version 16.0 and IBM SPSS Statistics version 26.

## Results

The number of countries submitting *estimates of requirements* for morphine included within our sample, increased year on year between 1997 and 2017, from 176 in 1997 to 187 in 2017 [Table 1]. More people died from cancer year on year, increasing overall needs for morphine. Total *calculated needs* for morphine for pain in advanced cancer, increased at each time point accordingly, from approximately five million in 1997, to seven million people in 2017. Total countries' *estimates of requirement* for morphine increased substantially, from 25,526kg in 2007 to 311,105kg in 2012, reducing slightly to 300,192kg in 2017.

In 1997, global *estimates of requirements* for morphine could feasibly meet only 86% of calculated global needs for people with advanced cancer (total *estimated requirement*/*calculated needs* for morphine*100). However, since 1997, availability of morphine has increased substantially with *estimates of requirements* sufficient to drastically exceed *calculated needs* for people with advanced cancer in 2002 (846%), 2007 (753%), 2012 (800%) and 2017 (701%).

The total number and proportion of High-income Countries (HICs) increased in our sample, although the proportion of world population in HICs did not. Total *estimate of requirements* for morphine accounted for in HICs increased from 81% (20,795/25,526*100) in 1997 to 85% (255,012/300,19,*100) in 2017. However, the proportion of total *calculated need* in HICs reduced in this time period, indicating that inequality in availability of morphine increased between income groups between 1997 and 2017. Increased gaps between proportion of calculated need for morphine and proportion of world *estimate of requirement* increased in Lower Middle-Income Countries (LMICs) and Upper Middle-Income Countries (UMICs).

For all country groups, fluctuations are present in population, *calculated needs* for morphine and *estimates of requirement* between time-points, notably in low-income and upper-middle income countries. Fluctuations are accounted for both by changes to individual countries' *estimate of requirement* and differences in countries included within each income group, either due to some countries not submitting *estimates of requirements* for individual years, or movement between income group of countries.

Fig 3 shows the proportion of total countries submitting *estimates of requirements* which could feasibly meet and exceed *calculated needs* for morphine of people with advanced cancer, increased: 1997 (28/176, 16%), 2002 (43/181, 24%), 2007 (42/186, 23%), 53/186, 28%), 2017 (57/187, 30%). In such countries, *calculated need* for morphine for people with advanced cancer could feasibly be met, with additional morphine supplies available for other clinical indications.

Whilst this shows that availability of morphine is increasing worldwide, the proportion of countries submitting *estimates of requirements* which did not even have the possibility of meeting the *calculated needs* of people with advanced cancer, reduced only from 84% in 1997, to 70% in 2017.

In countries submitting *estimates of requirements* sufficient to meet ≤100% of *calculated needs*, our analysis suggests that at least 1.75M people died from cancer in 2017 with zero chance of receiving INCB-recommended morphine analgesia. Put another way, in 2017, countries submitting *estimates of requirements* sufficient to meet ≤100% of *calculated needs*, submitted *estimates of requirements* which could, in a best case scenario, meet only 56% of *calculated needs*.

In 1997, 16 countries submitted *estimates of requirements*, but did not list any morphine, meaning that 0% of calculated needs could possibly be met [Fig 4]. This number reduced to

**Table 1. Total included countries, proportion of world population, cancer deaths and calculated totals of people who died from advanced cancer requiring morphine, and *estimates of requirements* by World Bank Income Group (1997–2017).**

| | | Year analysed | | | | |
|---|---|---|---|---|---|---|
| | | **1997** | **2002** | **2007** | **2012** | **2017** |
| **Total included countries (n)** | | 176 | 181 | 186 | 186 | 187 |
| Population | | 5.73 billion | 6.23 billion | 6.66 billion | 7.08 billion | 7.50 billion |
| People who died from cancer | | 6,142,563 | 6,747,754 | 7,223,673 | 8,003,555 | 8,811,563 |
| People requiring morphine (*calculated need*)* | | 4,914,050 | 5,398,203 | 5,778,938 | 6,402,844 | 7,049,250 |
| Estimated morphine needs (kg) | | 29,853 | 32,794 | 35,107 | 38,897 | 42,824 |
| Total *estimate of requirement* | | 25,526 | 277,574 | 264,246 | 311,105 | 300,192 |
| By level of World Bank country income category | *Low-Income Countries* | *54* | *61* | *49* | *37* | *35* |
| | % of total countries | 31 | 34 | 23 | 20 | 19 |
| | Total population | 1,951,847,968 | 2,529,922,992 | 1,276,933,000 | 847,078,000 | 718,489,000 |
| | % of world population | 34 | 41 | 19 | 12 | 10 |
| | People who died from cancer | 1,001,365 | 1,327,893 | 725,383 | 458,530 | 344,488 |
| | People requiring morphine (*calculated need*)* | 801,092 | 1,062,314 | 580,306 | 366,824 | 275,590 |
| | Morphine needs^ | 4,867 | 6,454 | 3,525 | 2,228 | 1,674 |
| | *Estimate of requirement* (kg) | 470 | 10,832** | 478 | 414 | 655 |
| | *Lower middle-income countries* | *54* | *52* | *51* | *45* | *43* |
| | % of total countries | 31 | 29 | 27 | 24 | 23 |
| | Total population | 2,309,154,032 | 2,434,515,064 | 3,484,886,944 | 2,542,306,960 | 2,961,446,984 |
| | % of world population | 40 | 39 | 52 | 36 | 39 |
| | People who died from cancer | 2,558,249 | 2,929,128 | 3,121,798 | 1,561,385 | 1,960,931 |
| | People requiring morphine (*calculated need*)* | 2,046,599 | 2,343,302 | 2,497,438 | 1,249,108 | 1,568,745 |
| | Morphine needs^ | 12,433 | 14,236 | 15,172 | 7,588 | 9,530 |
| | *Estimate of requirement* (kg) | 3,189 | 32,698 | 22,042 | 10,336 | 15,921 |
| | *Upper middle-income countries* | *30* | *29* | *40* | *52* | *54* |
| | % of total countries | 17 | 16 | 22 | 28 | 29 |
| | Total population | 574,424,992 | 330,804,000 | 870,549,000 | 2,426,880,960 | 2,613,612,958 |
| | % of world population | 10 | 5 | 13 | 34 | 35 |
| | People who died from cancer | 592,732 | 396,495 | 1,103,314 | 3,180,085 | 3,786,096 |
| | People requiring morphine (*calculated need*)* | 474,186 | 317,196 | 882,651 | 2,544,068 | 3,028,877 |
| | Morphine needs^ | 2,881 | 1,927 | 5,362 | 15,455 | 18,400 |
| | *Estimate of requirement* (kg) | 1,072 | 15,323 | 9,328 | 42,429 | 28,603 |
| | *High-income countries* | *35* | *39* | *46* | *52* | *55* |
| | % of total countries | 20 | 22 | 15 | 28 | 29 |
| | Total population | 899,514,984 | 931,338,008 | 1,023,183,000 | 1,265,167,000 | 1,209,952,992 |
| | % of world population | 16 | 15 | 15 | 18 | 16 |
| | People who died from cancer | 1,990,217 | 2,094,238 | 2,273,178 | 2,803,555 | 2,720,048 |
| | People requiring morphine (*calculated need*)* | 1,592,174 | 1,675,390 | 1818542 | 2,242,844 | 2,176,038 |
| | Morphine needs^ | 9,672 | 10,178 | 11,048 | 13,625 | 13,219 |
| | *Estimate of requirement* (kg) | 20,795 | 218,721 | 232,398 | 257,926 | 255,012 |

*Calculated at 80% of people who died from cancer;

^calculated as number of people x 67.5 mg per day x 90 days reported in kilograms

** A large spike in *estimate of requirement* in this year is accounted for by a large increase in a single country's (India) *estimate of requirement* for morphine in this year.

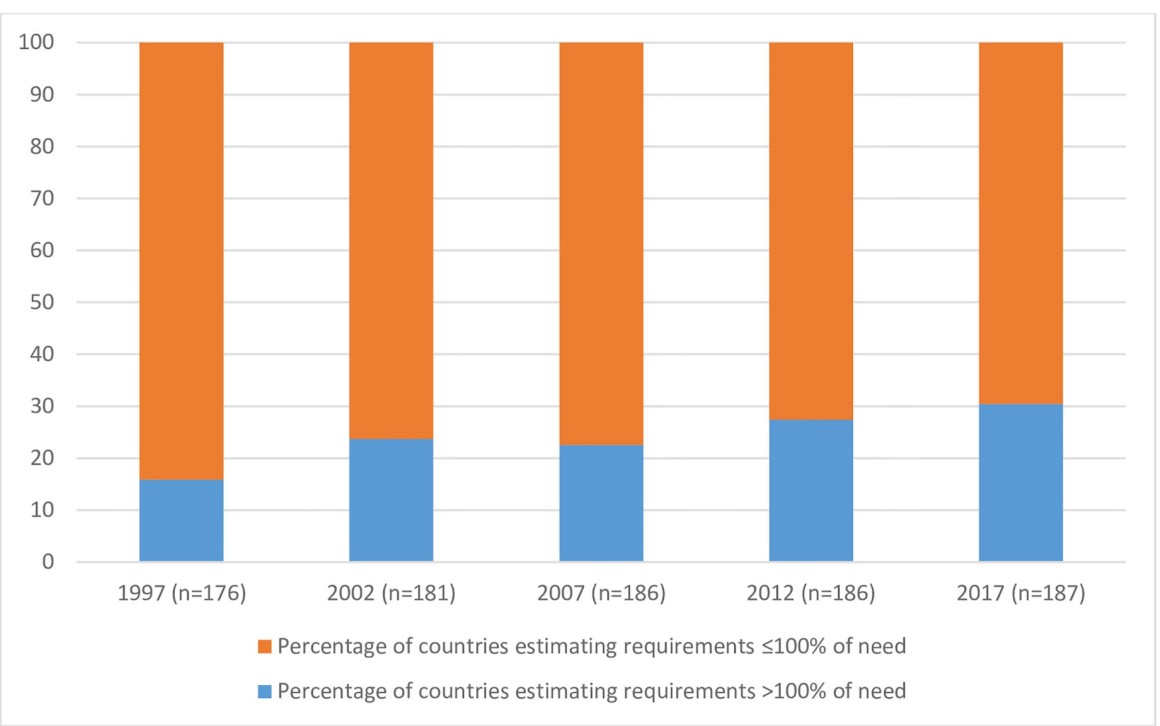

**Fig 3. Proportion of included countries submitting *estimates of requirements* which could feasibly meet > or ≤ 100% of calculated need for morphine of people with advanced cancer, 1997–2017.**

two countries by 2017. However, a high proportion of countries in all income settings submitted *estimates of requirements* which could feasibly meet only a small proportion of *calculated needs*. The proportion and distribution of countries which report *estimates of requirements* which could feasibly meet only a fraction of *calculated needs* do not appear to change materially over a 20 year period.

The proportion of countries which submitted *estimates of requirements* sufficient to meet >100% of *calculated needs* increased from 16% in 1997 to 30% in 2017. This indicates that an increasing proportion of world countries are submitting *estimates of requirements* for morphine which exceeds quantities necessary to treat people who die from advanced cancer at INCB-recommended dosage and duration, with additional morphine then available for other approved clinical and scientific indications such as post-operative and trauma analgesia.

## Income group and morphine availability

Countries submitting *estimates of requirements* for morphine insufficient for meeting 100% of *calculated needs* are present in all income groups [Fig 5]. The number of countries within income groups changes within our sample. However, the proportion of countries within each income group submitting *estimates of requirements* which could feasibly meet *calculated needs* for morphine for cancer-related pain relief, with additional capacity to treat others requiring morphine, changes little between 1997 and 2017 in any income group.

**Consumption of morphine (morphine feasibly accessed by people with advanced cancer).** Of countries reporting both *estimates of requirements* and *consumption* for an included year, patterns of morphine *consumption* worldwide are consequently highly consistent with

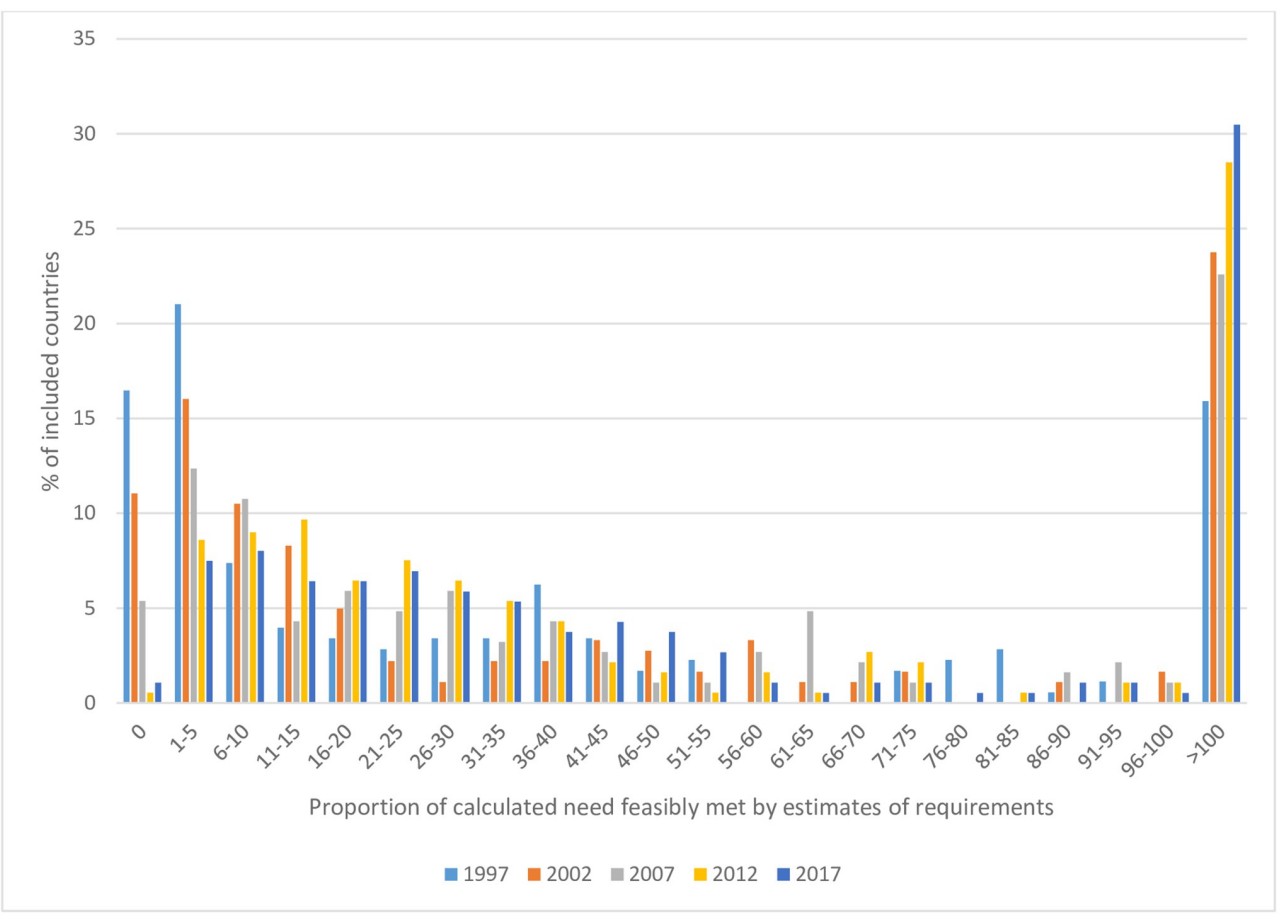

**Fig 4. Proportion of calculated need feasibly met by *estimates of requirements* in 1997 (n = 176), 2002 (n = 181), 2007 (n = 186), 2012 (n = 186) and 2017 (n = 187).**

those identified for *estimates of requirements* [S1–S3 Figs], with lower proportions of calculated need feasibly met by *consumption* than *estimates of requirements* [Table 2].

For each stage of analysis, proportions of *calculated needs* for morphine *feasibly treatable* from reports of *consumption* were lower than those reported for *estimates of requirements* indicating that *estimates of requirements* contextualise future consumption. Very few individual countries reported *consumption* levels that exceeded *estimates of requirements* (1997,22/143 (15%), 2002, 14/156 (9%), 2007, 12/156 (8%), 2012, 7/144 (5%), 2017, 15/153 (10%)). This indicates that few countries avail of flexibilities within the *estimates system* to increase *consumption* beyond *estimates of requirements* and existing stocks and supplies.

## Discussion

Worldwide under-consumption of morphine is well reported, notably by the Lancet Commission on Pain and Palliative Care [4]. However, the role of *estimates of requirements* in relation to reported under-*consumption* has not been previously explored empirically. Although morphine is indicated for a number of clinical problems (e.g. trauma), the INCB only publishes guidance on how to calculate *estimates of requirements* for people who died with moderate-severe cancer-related pain (80% of people who die from cancer will require 67.5 mg per day for 90 days). We therefore present a 'best case scenario' for the prospects of people who die

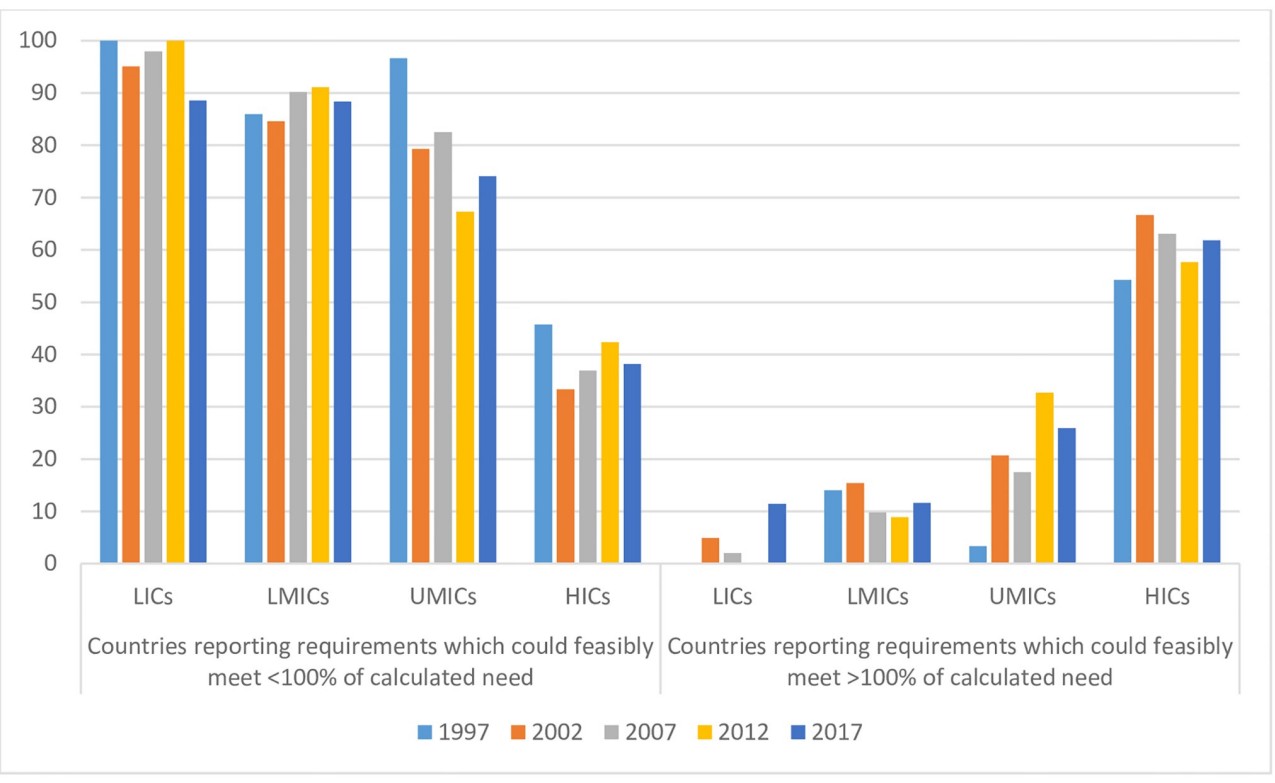

**Fig 5. Proportion of countries within each income group submitting *estimates of requirements* able to meet ≤ or > 100% of calculated need by year.**

from advanced cancer receiving morphine over time. Legitimate needs for morphine for other medical purposes should be considered as additional to our calculations.

Our analysis indicates most countries worldwide systematically submit *estimates of requirements* for morphine guaranteed not to meet the needs of people who die from cancer and that *estimates of requirements* contextualise inadequate *consumption*. This means countries submitting *estimates of requirements* insufficient to meet clinical needs are guaranteed to make inadequate morphine available for prescribing in that year.

We identify startling stability over twenty years in proportions of countries submitting inadequate *estimates of requirements*. In a twenty year period, the proportion of countries submitting *estimates of requirements* which could feasibly meet >100% of *calculated need* for people who die from cancer increased only from 16% to 30% between 1997 and 2017. In every included year within our sample globally, >70% of countries were guaranteed to provide inadequate availability of morphine for people with moderate-severe pain in advanced cancer.

Our model shows that the proportion of *calculated need* for morphine of people who died from advanced cancer feasibly met by *estimates of requirements* rose from 86% in 1997 to 701% in 2017. This means that some countries have increased the proportions of *calculated need* feasibly met by *estimates of requirements*, with additional morphine for people with other clinical conditions. However, the vast majority of increased *estimates of requirements* has occurred in HICs and global inequity has not reduced. In HICs, between 1997 and 2017, *calculated need* for morphine increased from 9,672kg to 13,219kg, however, *estimates of requirements* increased from 20,795kg to 255,012kg. This huge discrepancy indicates potential overuse of morphine in HICs, that a significant amount of requirements are for indications

**Table 2. Total included countries reporting both *estimates of requirements* and *consumption*, proportion of world population, cancer deaths and proportions of calculated need feasibly met by countries' *estimates of requirements* and *consumption*, by World Bank Income Group (1997–2017).**

| | | Year analysed | | | | |
|---|---|---|---|---|---|---|
| | | **1997** | **2002** | **2007** | **2012** | **2017** |
| Total included countries reporting both *Estimates of requirements* and *Consumption* (n) | | 143 | 156 | 156 | 144 | 153 |
| Total population | | 5.51 billion | 6.09 billion | 6.42 billion | 6.73 billion | 7,01 billion |
| Number of people who died from cancer | | 5,978,734 | 6,674,669 | 7,021,569 | 7,780,895 | 8,549,962 |
| Number of people requiring morphine (*calculated need*)* | | 4,782,987 | 5,339,736 | 5,617,255 | 6,224,716 | 6,839,969 |
| Total *Estimate of requirement* (kg) | | 25,486 | 277,494 | 263,305 | 310,843 | 299,638 |
| Total *Consumption* | | 17,850 | 27,374 | 39,369 | 45,265 | 43,170 |
| Proportion of *calculated need feasibly treatable* using *Estimates of requirements* | | 88 | 855 | 772 | 822 | 721 |
| Proportion of *calculated need feasibly treatable* using *Consumption* | | 61 | 84 | 115 | 120 | 104 |
| By level of World Bank country income category | *Low-income countries* | 32 | 49 | 38 | 20 | 19 |
| | Total population | 1,775,004,968 | 2,438,094,992 | 1,177,084,000 | 583,716,000 | 389,045,000 |
| | Number of people who died from cancer | 893,778 | 1,281,473 | 675,380 | 304,268 | 179,011 |
| | Number of people requiring morphine (*calculated need*) | 715,023 | 1,025,178 | 540,304 | 243,414 | 143,209 |
| | Total *Estimate of requirement* (kg) | 450 | 10,823 | 429 | 304 | 327 |
| | Total *Consumption* | 31 | 70 | 104 | 79 | 162 |
| | Proportion of *calculated need feasibly treatable* using *Estimates of requirements* | 10 | 174 | 13 | 21 | 38 |
| | Proportion of *calculated need feasibly treatable* using *Consumption* | 1 | 1 | 3 | 5 | 19 |
| | *Lower middle-income countries* | 48 | 45 | 40 | 32 | 35 |
| | Total population | 2,260,732,032 | 2,392,636,064 | 3,354,455,944 | 2,502,958,960 | 2,859,820,984 |
| | Number of people who died from cancer | 2,504,071 | 2,905,174 | 2,981,286 | 1,532,230 | 1,911,678 |
| | Number of people requiring morphine (*calculated need*) | 2,003,257 | 2,324,139 | 2,385,029 | 1,225,784 | 1,529,343 |
| | Total *Estimate of requirement* (kg) | 3,171 | 32,633 | 21,172 | 10,231 | 15,796 |
| | Total *Consumption* | 728 | 1,380 | 1,514 | 338 | 548 |
| | Proportion of *calculated need feasibly treatable* using *Estimates of requirements* | 26 | 231 | 146 | 137 | 170 |
| | Proportion of *calculated need feasibly treatable* using *Consumption* | 6 | 10 | 10 | 5 | 6 |
| | *Upper middle-income countries* | 28 | 25 | 36 | 43 | 45 |
| | Total population | 572,023,992 | 327,907,000 | 864,609,000 | 2,378,196,960 | 2,550,288,952 |
| | Number of people who died from cancer | 2,504,071 | 394,029 | 1,092,793 | 3,141,650 | 3,739,336 |
| | Number of people requiring morphine (*calculated need*) | 472,534 | 315,223 | 874,234 | 2,513,320 | 2,991,469 |
| | Total *Estimate of requirement* (kg) | 1,070 | 15,320 | 9308 | 42,385 | 28,503 |
| | Total *Consumption* | 918 | 577 | 1537 | 2,630 | 3,621 |
| | Proportion of *calculated need feasibly treatable* using *Estimates of requirements* | 37 | 800 | 175 | 278 | 157 |
| | Proportion of *calculated need feasibly treatable* using *Consumption* | 32 | 30 | 29 | 17 | 20 |
| | *High-income countries* | 35 | 37 | 42 | 49 | 54 |
| | Total population | 899,514,984 | 931,202,008 | 1,021,932,000 | 1,264,728,000 | 1,209,857,992 |
| | Number of people who died from cancer | 1,990,218 | 2,093,993 | 2,272,110 | 2,802,747 | 2,719,935 |
| | Number of people requiring morphine (*calculated need*) | 1,592,174 | 1,675,195 | 1,817,688 | 2,242,198 | 2,175,959 |
| | Total *Estimate of requirement* (kg) | 20,795 | 218,719 | 232,396 | 257,922 | 255,012 |
| | Total *Consumption* | 16,173 | 25,346 | 36,214 | 42,218 | 38,839 |
| | Proportion of *calculated need feasibly treatable* using *Estimates of requirements* | 215 | 2,149 | 2,105 | 1,894 | 1,929 |
| | Proportion of *calculated need feasibly treatable* using *Consumption* | 167 | 249 | 328 | 310 | 294 |

other than cancer pain, or, that the formula for calculating needs of people who die from cancer is a vast underestimation.

Of the three approaches to calculating country *estimates of requirements* recommended by the INCB [Box 1], the apparent stability of inadequate *estimates of requirements* makes it likely that most countries use the consumption-based method, which uses previous consumption levels to estimate future *estimates of requirements*. This means that historic (inadequate) consumption is used to estimate future *requirements*, systematising a cycle of inadequate availability and utilisation of morphine [12]. We conclude that for the majority of countries worldwide, either the methods of *estimating requirements* relative to need have not improved in the last twenty years or no progress has been made in terms of system capacity to ensure safety of the supply chain.

Inadequate *estimates of requirements* contextualise endemic under-consumption of morphine in most countries worldwide. Between 1997 and 2017, countries reporting consumption sufficient to meet >100 of *calculated needs* of people who died from cancer, increased only from 13% in 1997, to 19% in 2017. This means in 2017, 81% of countries reported consumption levels guaranteed to provide inadequate availability of morphine for people with moderate-severe pain in advanced cancer.

The stability in global inequity of access to pain relief suggests that the *Estimates system* is not succeeding in its purpose as a pre-requisite of ensuring availability [13]. The lack of apparent progress in improving proportions of *calculated need* met by *estimates of requirements* and *consumption* over 20 years should be of concern to patients' advocates, clinicians and policy makers. Of further concern, there is no apparent immediate increase in proportions of need met, following the commitment of countries to ensure access to pain relief as part of the 2014 World Health Assembly Resolution strengthen palliative care as a component of comprehensive health care.

Since 2016, the INCB has supported countries in improving their *estimates of requirements* through the INCB Learning program [19]. It will be important to evaluate this program as to its usefulness in improving the functioning of the *estimates system*. Publication of *calculated needs* alongside country *requirements* would highlight gaps between need and provision and both drive countries accountability to meet the pain treatment needs of their populations and guard against overestimation, which risks inappropriate over access.

If countries take measures to improve appropriate access to morphine, increased *estimates of requirements* will be the first indication of potential changes in adequacy of consumption. Without reform, trends of inadequate *estimation of requirements* identified by our study, contextualising under (and over) access to morphine appear likely to continue, leaving millions suffering avoidably.

## Study limitations

Our data do not allow us to infer that in countries estimating >100% of *calculated needs*, all people with advanced cancer will receive morphine appropriately. However, absolute conclusions can be reached for countries reporting *estimates of requirements* which feasibly meet ≤100% of *calculated needs*.

Nevertheless, our analysis uses assumptions which introduce systematic over- and under-calculation of needs for morphine of people who died from cancer. Morphine is indicated for several clinical problems (e.g. trauma, [20] post-operative analgesia) in addition to people who die from cancer [21]. However, given no data are available as to the purposes for which *estimates of requirements* are made nor consumed, our analysis assumes that all *estimates of requirements* and country consumption are for the purposes of treating people who die from

advanced cancer. Additionally, much morphine worldwide is used in the manufacture of other medicines which means that clinical availability will be over-estimated. Additionally, as it is defined by 1961 Convention, a drug shall be regarded as "consumed" when it has been supplied to any person or enterprise for retail distribution, medical use or scientific research–meaning again, that we overcalculate proportions of need feasibly met by consumption.

Population estimates of people who died from cancer are also likely to systematically under-estimate mortality, due to under-registration of cancer as cause of death. By using Income Group as our unit of analysis, our approach also does not identify individual countries who may have either increased (or decreased) proportions of need met by *requirements*/ consumption.

Additionally, our use of a repeated time-series approach means that total countries included at each time point differ by year. This explains some of the fluctuation in our outcome variables presented in Table 1. For example, there is notable fluctuation in countries' population, *calculated need* and *estimate of requirements* in low-income countries, between 1997, 2002 and 2007. This is accounted for by an increase of six countries in the sample from 1997 to 2002 and a significant increase in India's estimates of requirement between the dates, from 400kg in 1997, to 10,000kg in 2002. India was included as a lower-middle income country by 2002, where after total estimate of requirement in low-income countries dropped to similar levels seen in 1997 and remained consistent.

Nevertheless, given the systematic biases mentioned and the clear trends of inadequate *estimations of requirement* and reported consumption, we offer a starting point for consideration of estimates of *requirements* and *consumption*, in context of *calculated needs* to drive the accountability of global regulation and countries worldwide.

## Supporting information

**S1 Fig. Proportion of included countries reporting consumption which could feasibly meet > or ≤ 100% of calculated need for morphine of people with advanced cancer, 1997–2017.**
(DOCX)

**S2 Fig. Proportion of included countries within income groups reporting consumption which could feasibly meet or ≤ or >100% of calculated needs by year, 1997–2017.**
(DOCX)

**S3 Fig. Proportion of calculated need feasibly met by consumption, 1997–2017.**
(DOCX)

## Acknowledgments

We are grateful to the INCB for responding to our request and providing estimated requirements and consumption data for 1997.

## Author Contributions

**Conceptualization:** Joseph Clark, Miriam J. Johnson, David C. Currow.

**Data curation:** Joseph Clark, Lucia Crowther.

**Formal analysis:** Joseph Clark, Lucia Crowther, Miriam J. Johnson, Christina Ramsenthaler, David C. Currow.

**Investigation:** Joseph Clark, Lucia Crowther, Miriam J. Johnson, David C. Currow.

**Methodology:** Joseph Clark, Lucia Crowther, Miriam J. Johnson, Christina Ramsenthaler, David C. Currow.

**Project administration:** Joseph Clark, Lucia Crowther, David C. Currow.

**Resources:** Joseph Clark.

**Software:** Joseph Clark.

**Supervision:** Miriam J. Johnson, Christina Ramsenthaler, David C. Currow.

**Validation:** Miriam J. Johnson, Christina Ramsenthaler, David C. Currow.

**Writing – original draft:** Joseph Clark, Lucia Crowther, Miriam J. Johnson, David C. Currow.

**Writing – review & editing:** Joseph Clark, Lucia Crowther, Miriam J. Johnson, Christina Ramsenthaler, David C. Currow.

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
