## [Decision Letter · Decision Letter 0]

20 Apr 2022

PGPH-D-21-01017

Calculating worldwide needs for morphine for pain in advanced cancer and proportions theoretically met by country estimates of requirements and consumption. Retrospective, time-series analysis (1997-2017)

Dear Dr. Clark,

Thank you for submitting your manuscript to PLOS Global Public Health. After careful consideration, we feel that it has merit but does not fully meet PLOS Global Public Health’s publication criteria as it currently stands. Therefore, we invite you to submit a revised version of the manuscript that addresses the points raised during the review process.

This study adds immense value to a rather underrepresented aspect of global public health in oncology. I congratulate the authors for their commendable work. Look forward to your submission in response to the revisions suggested.

We look forward to receiving your revised manuscript.

Kind regards,

Nikita Mehra, M.D., DM.,

Academic Editor

Journal Requirements:

1. Your co-authors, Miriam J Johnson (miriam.johnson@hyms.ac.uk), Christina Ramsenthaler (christina.ramsenthaler@hyms.ac.uk), and David C Currow (david.currow@uts.edu.au), have not confirmed authorship of the manuscript. We have resent them the authorship confirmation email; however please check that the above email address for them is correct and follow up personally to ensure they confirm. Please note that we cannot pass your manuscript to Production until we have received confirmations from all co-authors.

Just in case your co-authors are having difficulty confirming their authorship, you may advise them to send us an email at globalpubhealth@plos.org and we will confirm their authorship on the authors' behalf.

2. Please provide a detailed Financial Disclosure statement. This is published with the article, therefore should be completed in full sentences and contain the exact wording you wish to be published.

i) Please include all sources of funding (financial or material support) for your study. List the grants (with grant number) or organizations (with url) that supported your study, including funding received from your institution. 

ii). State the initials, alongside each funding source, of each author to receive each grant.

iii). State what role the funders took in the study. If the funders had no role in your study, please state: “The funders had no role in study design, data collection and analysis, decision to publish, or preparation of the manuscript.”

iv). If any authors received a salary from any of your funders, please state which authors and which funders.

3. Please update your Competing Interests statement. If you have no competing interests to declare, please state: “The authors have declared that no competing interests exist.”

4. Please note that your Data Availability Statement is currently missing the repository name and/or the DOI/accession number of each dataset OR a direct link to access each database. If your manuscript is accepted for publication, you will be asked to provide these details on a very short timeline. We therefore suggest that you provide this information now, though we will not hold up the peer review process if you are unable.

5. We do not publish any copyright or trademark symbols that usually accompany proprietary names, eg (R), (C), or TM (e.g. next to drug or reagent names). Therefore please remove all instances of trademark/copyright symbols throughout the text, including IBM® SPSS® on page 9.

**Comments to the Author**

1. Does this manuscript meet PLOS Global Public Health’s publication criteria? Is the manuscript technically sound, and do the data support the conclusions? The manuscript must describe methodologically and ethically rigorous research with conclusions that are appropriately drawn based on the data presented.

Reviewer #1: Yes

2. Has the statistical analysis been performed appropriately and rigorously?

Reviewer #1: Yes

3. Have the authors made all data underlying the findings in their manuscript fully available (please refer to the Data Availability Statement at the start of the manuscript PDF file)?

Reviewer #1: Yes

4. Is the manuscript presented in an intelligible fashion and written in standard English?

Reviewer #1: Yes

5. Review Comments to the Author

Reviewer #1: Overall

This is an important work and highlights a neglected issue in LMIC context. Overall paper is hard to read and follow as the terms need, requirement, theoretical availability and consumption are very hard to understand. Authors need to give clear definitions and use consistent language to make it understandable. It will useful to use only 2-3 key indicators to explain trends. The paper uses very important concepts from supply chain management which may not be well understood by medical / public health audience.

Specific comments

• Introduce the audience to key supply chain management concepts relevant to the paper such as forecasting, availability and consumption.

• Operational definitions in methods: Estimated need, estimated requirement, theoretical availability and consumption should be clearly defined in the methods section. Ensure use of consistent terminology in methods, results and tables for clear understanding.

• Box 2 is very useful. Connect the methods given in box to indicators. Estimated need, estimated requirement, theoretical availability –mention which is based on consumption and which is based on morbidity method.

• Table 1 : Estimated requirement for year 2002 for low and low middle income countries is very high compared to subsequent years. Explain in results and discussion.

• Table 1 : In the upper middle income countries, estimated requirement follows a fluctuating patterns. Explain in results and discussion.

• Line 201- 205 – Need vs requirement not clear

• Table 2 interpretation – line 231 -234 is not clear . Is this table adding to the overall conclusions? If not, it can be removed.

• Consumption of morphine Lines 239 -249 – This is an important para and should be supported by a table.

• Add a table which should compare consumption versus what countries had projected as the need and reported overall and if possible by country groups.

• Figure 4 and 5 – very hard to follow. Consider simplifying or explaining in the results

• Lines 265 – 278 Discussion section explains results. Consider moving it to results

6. PLOS authors have the option to publish the peer review history of their article (what does this mean?). If published, this will include your full peer review and any attached files.

**Do you want your identity to be public for this peer review?** For information about this choice, including consent withdrawal, please see our Privacy Policy.

Reviewer #1: No

---

## [Decision Letter · Decision Letter 1]

21 Jun 2022

Calculating worldwide needs for morphine for pain in advanced cancer and proportions feasibly met by country estimates of requirements and consumption. Retrospective, time-series analysis (1997-2017)

PGPH-D-21-01017R1

Dear Dr. Clark,

We are pleased to inform you that your manuscript 'Calculating worldwide needs for morphine for pain in advanced cancer and proportions feasibly met by country estimates of requirements and consumption. Retrospective, time-series analysis (1997-2017)' has been provisionally accepted for publication in PLOS Global Public Health.

Best regards,

Nikita Mehra, M.D., DM.,

Academic Editor

Reviewer Comments (if any, and for reference):

Reviewer's Responses to Questions

**Comments to the Author**

1. If the authors have adequately addressed your comments raised in a previous round of review and you feel that this manuscript is now acceptable for publication, you may indicate that here to bypass the “Comments to the Author” section, enter your conflict of interest statement in the “Confidential to Editor” section, and submit your "Accept" recommendation.

Reviewer #1: All comments have been addressed

2. Does this manuscript meet PLOS Global Public Health’s publication criteria? Is the manuscript technically sound, and do the data support the conclusions? The manuscript must describe methodologically and ethically rigorous research with conclusions that are appropriately drawn based on the data presented.

Reviewer #1: Yes

3. Has the statistical analysis been performed appropriately and rigorously?

Reviewer #1: Yes

4. Have the authors made all data underlying the findings in their manuscript fully available (please refer to the Data Availability Statement at the start of the manuscript PDF file)?

Reviewer #1: Yes

5. Is the manuscript presented in an intelligible fashion and written in standard English?

Reviewer #1: Yes

6. Review Comments to the Author

Reviewer #1: (No Response)

7. PLOS authors have the option to publish the peer review history of their article (what does this mean?). If published, this will include your full peer review and any attached files.

**Do you want your identity to be public for this peer review?** For information about this choice, including consent withdrawal, please see our Privacy Policy.

Reviewer #1: No
